# Nonthermal Radiation of the Extreme TeV Blazar 1ES 0229+200 from Electromagnetic Cascades on Infrared Photon Field

Timur Dzhatdoev [1],* , Vladimir Galkin [1,2] and Egor Podlesnyi [1,2]

1   Skobeltsyn Institute of Nuclear Physics (SINP MSU), Federal State Budget Educational Institution of Higher Education, M.V. Lomonosov Moscow State University, 1(2), Leninskie Gory, GSP-1, 119991 Moscow, Russia; v_i_galkin@mail.ru (V.G.); podlesnyi.ei14@physics.msu.ru (E.P.)
2   Department of Physics, Federal State Budget Educational Institution of Higher Education, M.V. Lomonosov Moscow State University, 1(2), Leninskie Gory, GSP-1, 119991 Moscow, Russia
*   Correspondence: timur1606@gmail.com

**Abstract:** Extreme TeV blazars (ETBs) are active galactic nuclei with jets presumably pointing towards the observer having their intrinsic (compensated for the effect of $\gamma$-ray absorption on extragalactic background light photons) spectral energy distributions (SEDs) peaked at an energy in excess of 1 TeV. These sources typically reveal relatively weak and slow variability as well as higher frequency of the low-energy SED peak compared to other classes of blazars. It proved to be exceedingly hard to incorporate all these peculiar properties of ETBs into the framework of conventional $\gamma$-ray emission models. ETB physics have recently attracted great attention in the astrophysical community, underlying the importance of the development of self-consistent ETB emission model(s). We propose a new scenario for the formation of X-ray and $\gamma$-ray spectra of ETBs assuming that electromagnetic cascades develop in the infrared photon field surrounding the central blazar engine. This scenario does not invoke compact fast-moving sources of radiation (so-called "blobs"), in agreement with the apparent absence of fast and strong variability of ETBs. For the case of the extreme TeV blazar 1ES 0229+200 we propose a specific emission model in the framework of the considered scenario. We demonstrate that this model allows to obtain a good fit to the measured SED of 1ES 0229+200.

**Keywords:** astroparticle physics; non-thermal radiation mechanisms; active galaxies; BL Lacertae objects: individual: 1ES 0229+200

## 1. Introduction

Blazars are active galactic nuclei with their jets pointing in the close direction to the observer's line of sight. The observable spectral energy distribution (SED = $E^2 dN/dE$, where $E$ is the observable photon energy and $N$ is the number of photon counts) of a typical blazar reveals two prominent peaks. As a rule, the lower-energy SED peak occupies the frequency range spanning from infrared to X-rays; the higher-energy SED peak falls into the $\gamma$-ray domain. Strong and fast variability, sometimes on timescales as short as minutes [1,2], is typical for most classes of blazars. This flaring emission is usually interpreted in a theoretical framework assuming particle acceleration and subsequent $\gamma$-ray production inside relativistic plasmoids [3] (usually called "blobs" [4,5]) propagating along the jets.

In 2006, the H.E.S.S. collaboration reported the discovery of $\gamma$-ray emission from the blazar 1ES 1101-232 at redshift $z = 0.186$. The SED of this source was measured up to the energy of about 3 TeV [6]. At such energies, $\gamma$ rays are subject to strong absorption on extragalactic background light (EBL) photons due to the $\gamma\gamma \to e^+e^-$ pair production (PP) process [7,8] (for the observable energy of 1 TeV and the source redshift of 0.1, the optical depth of the PP process $\approx 1$). Neglecting the effects arising from intergalactic electromagnetic cascades (these effects are discussed in Section 6 below), the intrinsic spectrum of 1ES 1101-232 (i.e., the spectrum of $\gamma$ rays escaping into the intergalactic medium) was reconstructed.

Remarkably, this intrinsic spectrum does not reveal a cutoff even at the highest observable energy (for instance, see Fig. 17 (top-left) of [9]).

Such blazars with intrinsic SEDs peaked at an energy $E_{ph} > 1$ TeV are called "extreme TeV blazars" (ETBs). ETBs have recently came into the spotlight of the topical research in astrophysics [10]. By the end of 2019 about ten ETBs were discovered [10]. However, it proved to be notoriously difficult to arrive at a satisfactory theoretical understanding of these sources. In the present paper we inquire into the nature of high energy ($E > 100$ MeV) $\gamma$ rays and X-rays ($E = 0.3$–$100$ keV) from extreme TeV blazars. In Section 2 we summarize the basic peculiar properties of ETBs which any model seeking to explain their emission must be able to reproduce. In Section 3 we briefly discuss the difficulties of the existing models.

In the rest of the paper we propose and discuss an emission model for the blazar 1ES 0229+200 ($z = 0.14$ [11] ). This source is convenient for the aims of the present study because its broadband spectrum was well measured from the infrared to the very high energy (VHE, $E > 100$ GeV) domain and, moreover, because the spectrum was made publicly available in Supplementary Information of [10]. In Section 4 we examine the low-energy part of the spectrum and speculate about the possible spectrum of thermal radiation inside 1ES 0229+200. In Section 5 we fit the X-ray and $\gamma$-ray part of the spectrum with semi-analytic templates for electromagnetic cascade spectra, including both the inverse Compton and the synchrotron components. This paper is meant to be the first in series of works aimed at the understanding the nature of extreme TeV blazars. Therefore, we leave some subjects for future study (see Section 6). Finally, we conclude in Section 7.

## 2. Peculiar Properties of Extreme TeV Blazars

Below we list some observational properties of extreme TeV blazars. ETB emission model building could be guided by the requirement that a successful model must reproduce these properties.

1.  By definition, the intrinsic SED of any ETB is peaked at an energy $E_{ph} > 1$ TeV (see Section 1). In particular, the reconstructed intrinsic spectrum of the blazar 1ES 0229+200 [12] does not show an evidence for a break or cutoff even at the highest accessible energy of 10 TeV [9].
2.  ETBs do not reveal fast (day-scale or even week-scale) or strong (flux change by an order of magnitude or more) $\gamma$-ray variability (the only known exception is a day-scale flare of 1ES 1218+304 [13]).
3.  At the same time, relatively weak and slow (if compared to other classes of blazars) $\gamma$-ray variability of ETBs was indeed detected. In particular, the VERITAS Collaboration finds that the blazar 1ES 0229+200 is variable on yearly timescales [12]. In the present paper we strive to explain the variability quasi-period as small as several months, at the same time explaining the absence of faster variability.
4.  The energy of the low-energy SED peak $E_{pl}$ for ETBs is usually relatively high compared to other classes of blazars and falls into the X-ray domain [10]; this could incur significant difficulties in some models as discussed in Section 3.

## 3. Models of Extreme TeV Blazar Emission and Their Difficulties

Below we list some models that were proposed to describe the multiwavelength emission of ETBs. We note that most of these models assume the existence of blobs inside the jets of ETBs, at odds with the absence of fast and strong variability of these sources. As well, many models require that the magnetic field energy density ($u_B$) is well below the energy density of particles ($u_p$): $K_u = u_p/u_B \gg 1$.

The $K_u \gg 1$ condition is very hard to meet. Indeed, in this case the accelerating particles quickly escape from the blob, stopping acceleration immediately (for a discussion of the $K_u \sim 1$ asumption see e.g., [14,15], p. 7). In addition, such systems are usually characterised by fast magnetic field amplification driven by the particle current on a timescale comparable to the particle acceleration timescale [16]. This process would lead to

a shift of the low-energy SED component towards higher energies, impairing the fit to the observed spectrum.

1.  In synchrotron self-Compton (SSC) models $\gamma$ rays are produced as a result of inverse Compton (IC, $e^-\gamma \rightarrow e^{-'}\gamma'$ or $e^+\gamma \rightarrow e^{+'}\gamma'$) scattering of accelerated electrons on synchrotron photons radiated by the same electron population. For the case of ETBs (assuming $\gamma$-ray production in blobs) these models require $K_u \gg 1$ as well as high values of minimal Lorentz factor of the primary electrons ($\gamma_{e-min} > 10^4$) (e.g., [12]). The last condition requires specific acceleration mechanism such as the Blandford-Znajek mechanism [17] that operates only in the immediate vicinity of the event horizon of the central black hole in the blazar. This mechanism does not chime well with the concept of particle acceleration inside a fast-moving blob that quickly escapes the central engine.

2.  A leptonic model presented in [18] assumes that a high value of $\gamma_{e-min}$ is due to a specific preacceleration process, namely, the transfer of energy from protons to electrons. However, it is not clear why the same process does not operate in jets of other (non-extreme) blazars. This model, as well, implies $K_u \gg 1$.

3.  Another leptonic model assuming the IC process on cosmic microwave background (CMB) photons was proposed in [19]. It still suffers from the $\gamma_{e-min}$ problem outlined above. In addition, the CMB energy density typically dominates very far from the central black hole, at the multi-parsec or even kiloparsec scale. Thus appears the problem of ultrarelativistic electron transport to such great distances without appreciable energy losses and/or the problem of accelerating such electrons very far from the central engine.

4.  Hadroleptonic models relying on production of $\gamma$ rays, electrons and positrons in blobs via photohadronic processes (e.g., [20]) usually require $K_u \gg 1$.

5.  The proton sychrotron model (e.g., [20]) requires the acceleration of primary protons up to $\sim 10^{20}$ eV. The maximum energy of synchrotron photon produced by the protons is $E_{sp-max} \sim 100$ GeV, and for electrons co-accelerated with the protons $E_{se-max} \sim 50$ MeV [21]. To produce hard (spectral index $< 2$) observable spectrum up to the energy of $E_{ph} = 10$ TeV, the bulk motion of a blob or a jet corresponding to the Doppler factor of $D > E_{ph}/E_{sp-max} = 100$ is required. In this case, the typical observable energy of synchrotron photons emitted by electrons would be $\sim E_{se-max}D \approx 5$ GeV, and not $\sim$1–10 keV typical for extreme blazars. Therefore, it is difficult to account for the low-energy SED component self-consistently.

For these reasons, a radically different approach to the ones proposed in the above-discussed works is justified when trying to understand the emission of extreme TeV blazars. As a first step of this understanding process, let us consider the nature of photon field on which the observed $\gamma$ rays could be produced.

## 4. The Low-Energy Part of the Spectrum

The low-energy part of the observed SED for the blazar 1ES 0229+200 is plotted in Figure 1. All data are taken from tables provided as Supplementary Material for [10]. X-ray data are from the Neil Gehrels Swift Observatory [22], X-ray Telescope (Swift-XRT) and the NuSTAR Observatory [23], according to the analysis of [24] (see red circles in their Figure 2). The data in the energy range of 0.05–10 eV are from the the Swift Ultra-violet Optical Telescope (Swift-UVOT) (for two epochs, see Supplementary Information of [10,24] for more details), the Wide-field Infrared Survey Explorer (WISE) satellite [25], the ASI Space Science Data Center (ASI-SSDC) (again, see [24] for more details), and [26] (see legend in Figure 1).

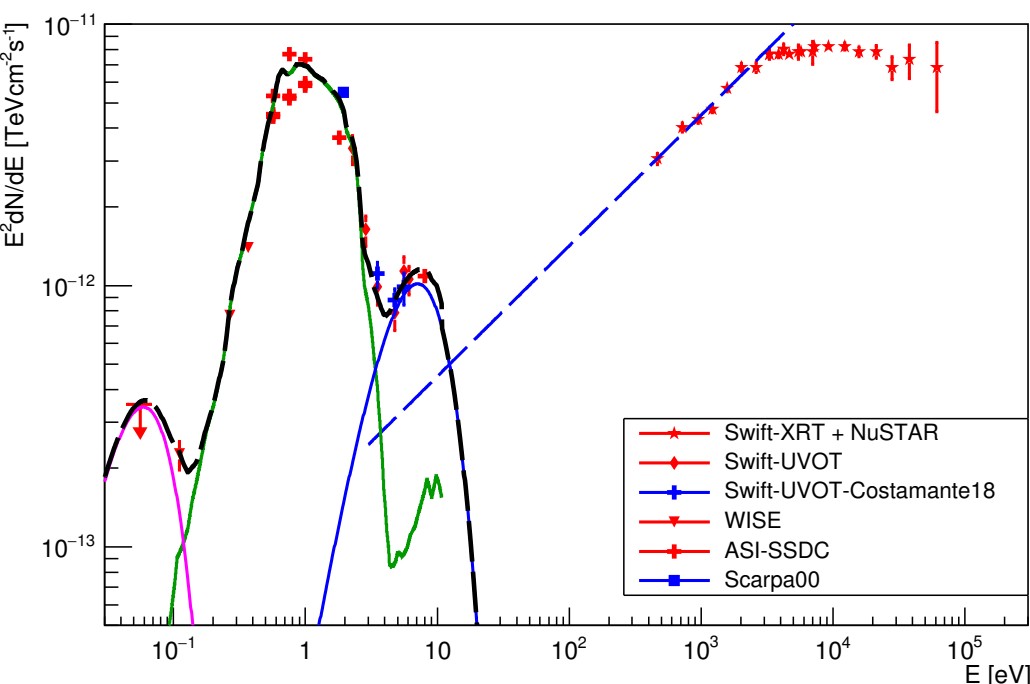

**Figure 1.** The low-energy part of the SED for 1ES 0229+200 observed with various telescopes. Red stars denote X-ray measurements, red and blue symbols of other shapes—optical, UV, and IR measurements. Green solid curve denotes the giant elliptical galaxy template of S98, blue solid curve— the blackbody component peaking in the UV energy range, magenta solid curve—the blackbody component peaking in the IR energy range, thick dashed black curve—the sum of the latter three components. Dashed blue line denotes a low-energy extrapolation of the X-ray component. Red arrow denotes an upper limit.

Below the hydrogen ionisation edge, the SED consists of three distinct bumps. The SED of the central bump is well fitted with the giant elliptical galaxy template of [27] (hereafter S98); indeed, the host galaxies for most ETBs are believed to be giant ellipticals [24]. However, in the ultraviolet (UV) and infrared (IR) ranges some residuals with respect to this template are evident; these residuals are well fitted with two blackbody components that have the temperatures of 175 K and $2.1 \times 10^4$ K and the luminosities of $4 \times 10^{43}$ erg/s and $1.2 \times 10^{44}$ erg/s, respectively.

The ultraviolet component may represent a stellar phenomenon [28]. In principle, some contribution to the UV component from an accretion flow is also possible (this was the case of the elliptical galaxy NGC 4552 [29]). It is believed that the inner part of the accretion flow in Bl Lacs is radiatively inefficient, with high temperature beyond the UV energy range. However, as discussed in [30], the outer part of the accretion flow could represent a standard Shakura-Sunyaev accretion disk [31].

Now let us discuss the infrared part of the spectrum in more details (see Figure 2). The template of S98 predicts a bump peaked at the energy of $\sim 10^{-2}$ eV (this component is probably due to radiation of dust), while observations reveal a more energetic component peaked at $\approx 6 \times 10^{-2}$ eV. The discrepancy in the peak energy and intensity of the IR component between the template of S98 and the observations may be in part due to the underestimation of the typical temperature of the dust and/or of the dust content in the star-forming regions.

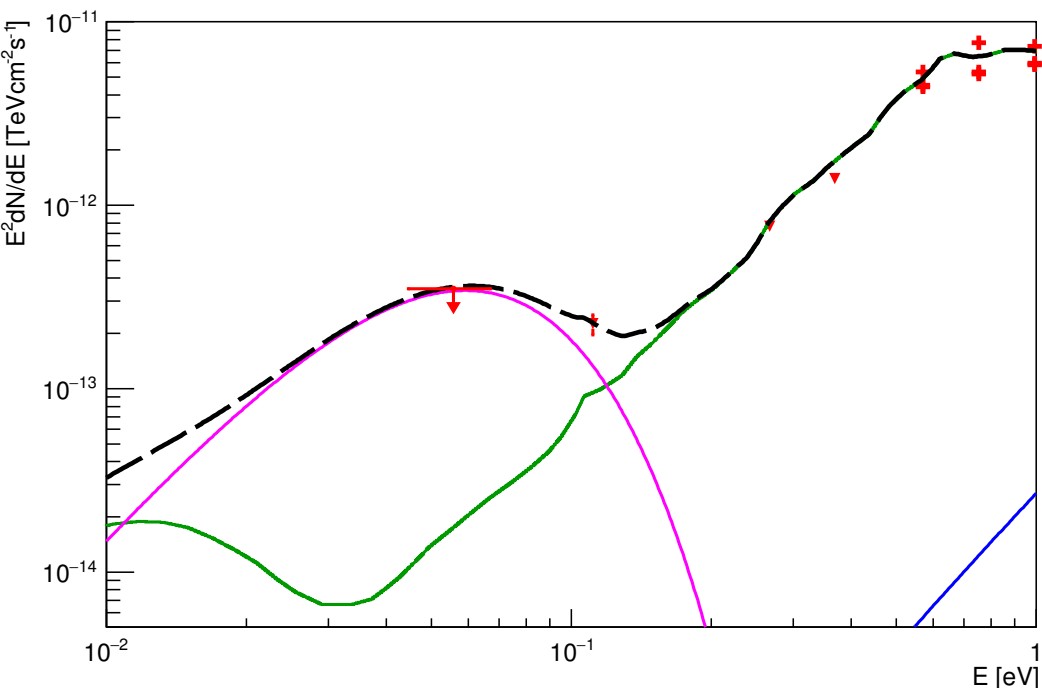

**Figure 2.** The infrared part of the SED for 1ES 0229+200. Colors are the same as in Figure 1.

However it is also possible that a part of the observed IR photons was produced near the central engine, at the spatial scale between ∼0.01 pc and 100 pc (the intensity of photon fields in the energy range $\leq 5 \times 10^{-2}$ eV in 1ES 0229+200 is poorly constrained with available observations). This IR photon field could serve as a target for high-energy electrons that could produce VHE $\gamma$ rays via the IC process. The same electrons could produce the X-ray component of the spectrum via the synchrotron process. Let us see what shape of the observable spectrum we could expect in the framework of this scenario. For simplicity we assume that the IR photon field is isotropic.

## 5. The High-Energy Part of the Spectrum

### 5.1. The Measured SED

The high-energy part of the observed SED for the blazar 1ES 0229+200, together with the X-ray component, is presented in Figure 3. The spectra obtained with imaging atmospheric Cherenkov telescopes were taken from [12] (VERITAS, red circles) and [32] (H.E.S.S., red triangles).

We have performed a Fermi-LAT [33] data analysis for 1ES 0229+200 over the time period from 4 August 2008 to 20 September 2020 and the energy range of 1 GeV–500 GeV following the standard recommendations for a point-like source spectral analysis issued by the Fermi-LAT collaboration, using the 4FGL catalog [34], the galactic background model gll_iem_v07, and the isotropic background model iso_P8R3_SOURCE_V2_v1. The Fermitools software version 1.2.23)[1] as well as the fermiPy package [35] (version 0.19.0)[2] were utilized for this analysis. For details concerning the H.E.S.S. and VERITAS data analysis the reader is referred to [12,32], respectively; for details concerning all datasets except the H.E.S.S., VERITAS, and Fermi-LAT ones—to [10]. We note that the observations with different instruments were not strictly simultaneous. However, as the source under study does not display a very strong or fast variability, in what follows we use the datasets resulting from these observations.

The Fermi-LAT SED of 1ES 0229+200 obtained by us is shown in Figure 3 as red squares.

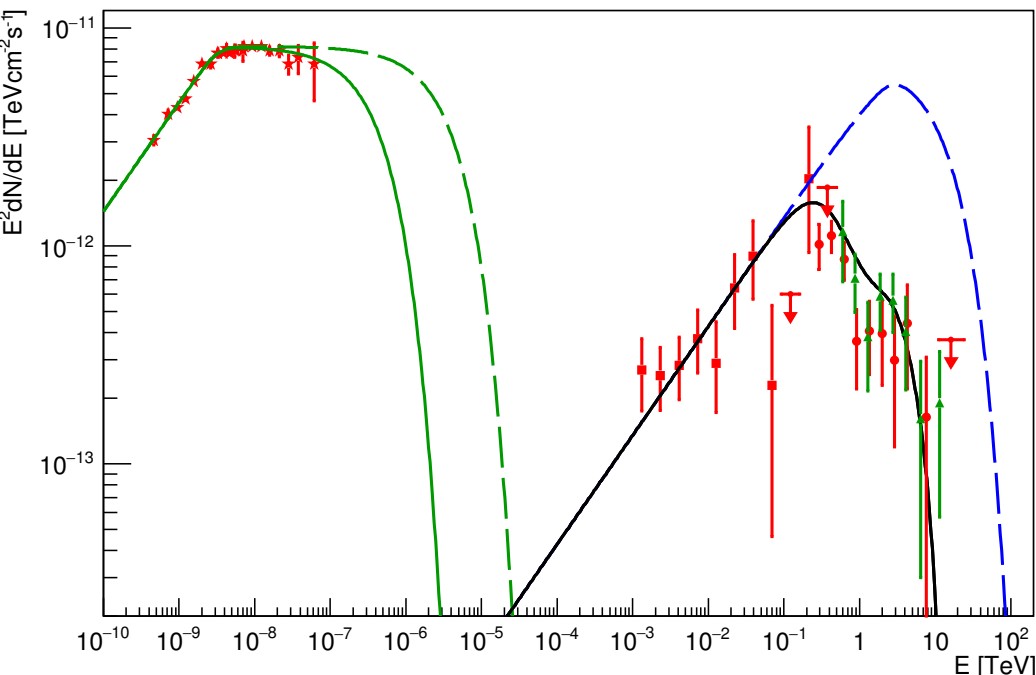

**Figure 3.** The high-energy part of the SED for 1ES 0229+200. Red symbols denote the measured SED: stars—X-ray measurements, squares—the Fermi-LAT SED, circles—VERITAS measurements, triangles—H.E.S.S. measurements. Arrows denote Fermi-LAT and VERITAS upper limits. Green curves denote fits to the X-ray component assuming different values of the maximum energy of the primary electrons: 100 TeV for solid curve and 300 TeV for dashed curve, dashed blue curve—the intrinsic $\gamma$-ray component but including the redshift effect, black curve—the observable $\gamma$-ray component including redshift and the EBL absorption.

*5.2. Qualitative Considerations*

Primary electrons could be accelerated near the central black hole of the blazar (typically at the distance less that several gravitational radii if counted from the event horizon) via the Blandford-Znajek process [17] up to at least 100 TeV (e.g., [36]). IC $\gamma$ rays produced by these electrons could create electron-positron pairs which, in turn, generate secondary (cascade) $\gamma$ rays. If the resulting electromagnetic cascade has more than two generations, its spectrum is usually almost independent on both the type of the primary particle (electron/$\gamma$ ray) and its energy. This property is called "cascade universality" [37].

Under the condition of universality, the spectrum of IC $\gamma$ rays has a definite shape: it is well-described with a $dN/dE \propto E^{-1.5}$ power law at relatively low energies, below a certain energy $E_x$; with a $dN/dE \propto E^{-2.0}$ dependence above $E_x$; and above the $\gamma$-ray horizon energy $E_a$ this spectrum reveals a cutoff caused by the PP process. Below we adopt an analytic representation for the IC component based on a smoothly broken power law functional form. For more details, the reader is referred to [37].

The same equations as those presented in [37] for the IC component could be written for the synchrotron component, except that the PP process does not matter for the synchrotron component due to much lower energies of synchrotron photons compared to IC photons. Indeed, both the synchrotron photon energy and the IC $\gamma$ ray energy in the Thomson regime are proportional to the square of the primary electron energy (e.g., [38]). For the case of photon field assumed below, the Thomson regime is valid up to the energy of $\sim$1 TeV. The deviation from the Thomson regime will be accounted for with the correction factor $f_{KN}$.

*5.3. The Model SED*

Assuming the cascade universality, we calculate the spectrum of cascade $\gamma$ rays and synchrotron photons as follows:

$$\frac{dN_\gamma}{dEdt} = \sum_{i=s,c} K_i E^{-\gamma_1} \left[1 + \left(\frac{E}{E_{xi}}\right)^\varepsilon\right]^{-\frac{\gamma_2-\gamma_1}{\varepsilon}} e^{-E/E_{mi}} e^{-\tau_{int}(E)}, \tag{1}$$

where $s$ stands for the synchrotron component, $c$—for the IC component; $K_i$ are the normalization factors for each component, $\gamma_1 = 1.5$, $\gamma_2 = 2.0$ [37]; $E_{xi}$ are the energies of the break of the power-law index value, $\varepsilon = 5$ is the parameter defining the sharpness of the break, $E_{mi}$ are the maximal energies of produced photons and $\gamma$ rays, and $\tau_{int}(E)$ is the value of the intrinsic optical depth for $\gamma$ rays (we note that for the synchrotron component $\tau_{int} = 0$).

$K_i$ are determined from fitting the measured SED. For the synchrotron component we estimate $E_{xs}$ and $E_{ms}$ as follows (see e.g., [39]):

$$E_{xs} = 60\,\text{keV}\left(\frac{B}{1\,\text{G}}\right)\left(\frac{E_{xe}}{1\,\text{TeV}}\right)^2, \tag{2}$$

$$E_{ms} = 60\,\text{keV}\left(\frac{B}{1\,\text{G}}\right)\left(\frac{E_{me}}{1\,\text{TeV}}\right)^2, \tag{3}$$

where $B = 0.9$ mG is the characteristic magnetic field strength, and $E_{me} = 100$ TeV is the primary electron spectrum cutoff energy. $E_{xe} = E_a/2$, where $E_a$ is the energy of the intrinsic $\gamma$-ray horizon:

$$E_a = E_t \left(\frac{E_b}{1\,\text{eV}}\right)^{-1}, \tag{4}$$

and $E_b = 6 \times 10^{-2}$ eV is the characteristic energy of the IR photons. We set $E_t = 1$ TeV.

The intrinsic optical depth for $\gamma$ rays (more precisely, its dependence on the energy $\tau_{int}(E)$) is usually calculated by integrating the interaction rate over the distance. The definition of the interaction rate could be found e.g., in [40]. The energy of the $\gamma$-ray horizon is the energy corresponding to $\tau_{int}(E) = 1$. $E_t$ is the effective threshold energy of a background photon interacting with a $\gamma$-ray via the PP process. For instance, the threshold for a 1 TeV $\gamma$-ray is $\sim 1$ eV; $E_t$ is inversely proportional to the $\gamma$-ray energy.

For the IC component:

$$E_{xc} = \frac{4}{3}\gamma_{xe}^2 E_b f_{KN}(E_{xe}), \tag{5}$$

where $\gamma_{xe} = E_{xe}/m_e$ is the electron Lorentz factor and $m_e$ is the electron mass [eV]. Using the approximation of [41] for the secondary $\gamma$-ray spectrum resulting from the IC process for the case of thermal photon field, we obtain $f_{KN}(E_{xe}) \approx 0.15$. We use a simple parametrization $\tau_{int}(E) = E/E_a$. In this work we are not interested in the precise shape of the intrinsic $\gamma\gamma$ absorption cutoff, because the intrinsic $\gamma$-ray horizon energy appears to be significantly higher than the EBL $\gamma$-ray horizon energy.

After thus calculating the spectrum of X-rays and $\gamma$ rays resulting from cascades, we apply the effects of redshift and EBL absorption assuming the EBL model of [42], and plot the resulting model curves in Figure 3 (solid green curve for the synchrotron component and solid black curve for the IC component). This fit was obtained without any formal optimization with a dedicated algorithm, but just by choosing reasonable values of input parameters and performing several tries. We note that the blue curve corresponds to the case of $E_{me} = 100$ TeV.

Finally, we try an option of $E_{me} = 300$ TeV (dashed green curve for the synchrotron component). The observable spectrum of the IC component is not sensitive to the value of $E_{me}$ because $E_{me}$ significantly exceeds the EBL $\gamma$-ray horizon energy.

For demonstration purposes, in Figure 4 we plot the model $\gamma$-ray spectrum multiplied by $E^{1.5}$ neglecting the absorption effect on the EBL. Furthermore, green dashed curve in

this figure does not include the effect of internal absorption either. Vertical black dashed lines correspond to the values of $E_{xc}$ (left line) and $E_a$ (right line). Remarkably, the ratio $E_a/E_{xc}$ is below ten. In more standard blazar models such a $\gamma$-ray spectrum would imply a very narrow spectrum of parent electrons, as was noted in [43]. In our model, however, this feature appears self-consistently.

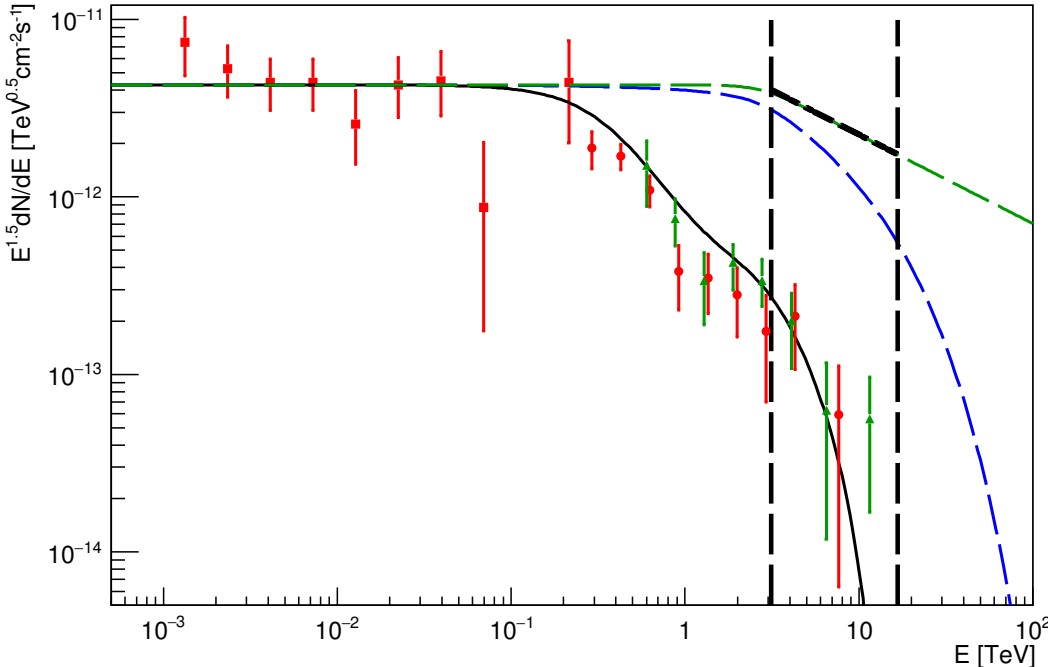

**Figure 4.** The high-energy part of the SED for 1ES 0229+200 and the corresponding model curves accounting the intrinsic absorption, but neglecting EBL absorption (blue curve) and without the account of both absorption effects (green curve). Black dashed segment denotes a $dN/dE \propto E^{-2}$ dependence. Upper limits are not shown.

*5.4. Basic and Auxiliary Parameters*

The proposed model of the spectral shape for the blazar 1ES 0229+200 has four basic parameters listed in Table 1. Besides these, the fifth parameter could be introduced in order to ensure proper absolute normalization to the SED measurements. The first three parameters appeared in the previous Subsection. We note that the physical meaning of $B$ is the strength of the turbulent component of magnetic field. The regular magnetic field along which the particles move could have a significantly higher strength than $B$. The fourth parameter defines the relative normalization of the IC and synchrotron components as follows:

$$K_{cs} = \frac{\int\limits_0^\infty E \frac{dN_c}{dE} dE}{\int\limits_0^\infty E \frac{dN_s}{dE} dE}, \tag{6}$$

where $dN_c/dE$ and $dN_s/dE$ are the spectra of the IC and synchrotron component, respectively. For the specific case considered in this paper, in the energy regions covered with observational data these spectra are almost independent from the $E_{me}$ value (see the previous Subsection). For these reasons, the shape of the model broadband spectrum is mostly defined by only three parameters: $E_b$, $B$, and $K_{cs}$.

**Table 1.** Basic parameters of the proposed model.

| Parameter | Meaning | Value | Units |
|:---:|:---:|:---:|:---:|
| $E_b$ | IR photon energy | $6 \times 10^{-2}$ | eV |
| $B$ | Magnetic field strength | 0.9 | mG |
| $E_{me}$ | Maximal energy of primary electron | 100 | TeV |
| $K_{cs}$ | IC SED peak dominance | 0.42 | |

In the previous Subsection we presented a fit to the measured SED assuming specific values of these parameters. We note that this solution is likely not unique and other sets of parameters could lead to a reasonable fit to the experimental data.

This fit included some auxiliary parameters that had their values fixed, but these values could in principle be obtained from *ab initio* calculations. Such direct and detailed calculations are presently underway; the corresponding results will be published elsewhere. The above-mentioned parameters include $(\gamma_1, \gamma_2)$ (the values of these follow from the theory developed in [37]), $E_t$ (its value is defined by the physics of the PP process), and $\varepsilon$. We note that these parameters are in fact defined by the values of $E_b$, $B$, and $K_{cs}$. Concerning the $f_{KN}$ parameter, we have already performed a calculation of its value, as explained in the previous Subsection.

## 6. Discussion

### 6.1. Explanation of Peculiar Properties of ETBs

As a summary of the above discussion, we present Table 2 which is called to explain, in a concise form, how the proposed model meets the expectations formulated in Section 2. Namely, when the IR photon field is sufficiently soft ($E_b \ll 1$ eV), it is possible to obtain the energy of the $\gamma$-ray SED peak $E_{ph} > 1$ TeV, because the PP effect does not lead to a strong cutoff in the cascade $\gamma$-ray spectrum below several TeV, and the Klein-Nishina cross section suppression [38] at such energies is still moderate. A sufficiently high energy of the primary electrons ($E_{me} \geq 100$ TeV) allows to produce multi-TeV cascade electrons which, in turn, radiate multi-keV synchrotron photons. Our model does not invoke relativistic blobs, naturally explaining the absence of strong or fast $\gamma$-ray variability. Finally, slow or weak variability could be explained by a variation in the accretion rate which, in turn, affects the properties of the primary electron spectrum (see the next Subsection for more details) and/or leads to variations in photon field properties which affect the spectrum of cascade $\gamma$ rays.

**Table 2.** Explanation of the peculiar properties of ETBs in the framework of the proposed model.

| Property | The Physical Reason |
|:---:|:---:|
| $E_{ph} > 1$ TeV | $E_b \ll 1$ eV $\rightarrow$ at 1 TeV $\tau_{int} \ll 1$, modest KN effects |
| $E_{pl} > 1$ keV | $E_{me} \geq 100$ TeV |
| Strong/fast var. absent | The absence of relativistic blobs |
| Slow/weak var. present | Variations of accretion rate/photon field |

### 6.2. The Origin of the Primary Electrons

As dissussed in Section 5.2, the primary electrons could have been accelerated in the vacuum gap near the central black hole via the Blandford-Znajek process [17,36,44–46]. The magnetic field in the vacuum gap $B_{gap}$ could be of order of 1 kG; therefore, the acceleration process could be accompanied by intense curvature radiation. For $B_{gap} > 1$ kG, the typical energy of the curvature photon radiated by $\sim$100 TeV electron falls in the gap between the X-ray and $\gamma$-ray parts of the spectrum of the blazar 1ES 0229+200 [47]. Future hard X-ray and MeV $\gamma$-ray observatories could possibly detect a distinct component from curvature radiation in the spectrum of 1ES 0229+200.

Another possible origin of ~100 TeV primary electrons is the Bethe-Heitler pair production process [48,49] on the IR photon field. In this case, the relevant primary proton energy is ~100 PeV–1 EeV. These protons could also have been accelerated via the Blandford-Znajek process [36,50,51].

A variation of the accretion rate induces a change of the magnetic field strength in the vacuum gap. The maximum total luminosity of the gap strongly depends on $B_{gap}$: $L_{gap} \propto B_{gap}^2$ [36,51]. Therefore, the change in the accretion rate could induce slow variability on the timescale of months or years. Finally, we note that a faster variability is not excluded altogether: discharges in the black hole magnetospheres are in principle possible leading to effects somewhat similar to lightning [52].

### 6.3. Intergalactic Electromagnetic Cascades

$\gamma$ rays escaping IR photon fields of the source initiate intergalactic electromagnetic cascades [53–55]. The observability of this intergalactic cascade signal from beamed sources strongly depends on the strength of the extragalactic magnetic field $B_{EGMF}$ and, under some circumstances, may also depend on its coherence length $\lambda_{EGMF}$. The existing constraints on the $(B_{EGMF}, \lambda_{EGMF})$ parameters are poor [56–62].

Above, we had neglected the intergalactic cascade contribution to the observable $\gamma$-ray spectrum. Indeed, if $B_{EGMF} > 10$ fG and $\lambda_{EGMF} > 100$ kpc, then this contribution from strongly beamed sources could be safely neglected [63,64]. Otherwise, the intergalactic cascade component could yield an observable signal. In 2017–2021, a series of studies was conducted in our research group discussing a possible contribution of the intergalactic cascade component to the observable spectra of extreme TeV blazars [9,65,66]. Preliminary estimates show that the proposed model allows a significant contribution of cascade $\gamma$ rays to the observable SED at low energies ($E < 1$ TeV). A detailed model of the observable spectrum of the blazar 1ES 0229+200 for the option of $B_{EGMF} \ll 10$ fG is currently in development and will be presented elsewhere.

### 6.4. Additional Remarks on the Model

Some $\gamma$-rays are inevitably produced via the SSC mechanism. However, the ratio of the number density for the IR photons to the X-ray photons is $K_{IR-X} \sim 10^3 / (\theta_{jet}^2)$, where $\theta_{jet}$ is the angular radius of the jet (we considered the X-ray photons from 0.3 to 60 keV in view of the fact that the cascade SED may have a low-energy cutoff [67]). For $\theta_{jet} = 0.1$ $K_{IR-X} \sim 10^5$; therefore, the SSC component is subdominant if both photon fields occupy the same volume (this is possible if the IR photons are produced by isotropized electrons that have left the jet and got trapped in magnetic fields near the jet's boundary).

Assuming that the UV component of photon field represents a stellar phenomenon on ~kpc scale (see Section 4), the pair production depth on this UV excess is small ($\ll 1$). In the present paper we do not invoke bulk motion of matter in blobs/jet to explain the non-thermal radiation spectrum of 1ES 0229+200. In this respect, our model is similar to the one presented in [36] for the case of the radiogalaxy M87.

The blackbody fits of the UV and IR components presented in Figure 1 are formal ones and do not have direct physical meaning. In particular, the IR component could be in part produced as the result of synchrotron radiation of electrons with relatively low energy that have escaped the jet.

### 7. Conclusions

In the present work we proposed a new scenario for the formation of X-ray and and $\gamma$-ray spectrum of the blazar 1ES 0229+200 assuming the acceleration of primary electrons up to the energy of ~100 TeV near the central black hole of the blazar with the subsequent development of electromagnetic cascades on the infrared photon fields surrounding the central engine. This scenario explains all known peculiar properties of extreme TeV blazars. Remarkably, the spectrum of 1ES 0229+200 is very similar to the spectrum of electromagnetic cascade in the universal regime. Moreover, we presented a

specific model that fits the measured SED of 1ES 0229+200 well. A somewhat modified version of this model could be applied to the case of other ETBs. Detailed calculations to this end are currently underway and will be published elsewhere.

**Author Contributions:** Conceptualization, T.D.; methodology, T.D. and E.P.; software, T.D., E.P. and V.G.; validation, T.D. and E.P.; formal analysis, T.D. and V.G.; investigation, T.D.; resources, T.D.; data curation, T.D. and V.G.; writing—original draft preparation, T.D.; writing—review and editing, T.D. and E.P.; visualization, T.D.; supervision, T.D.; project administration, T.D. and E.P.; funding acquisition, T.D. All authors have read and agreed to the published version of the manuscript.

**Funding:** The reported study was funded by RFBR, Russia, project number 20-32-70169.

**Institutional Review Board Statement:** Not applicable.

**Informed Consent Statement:** Not applicable.

**Data Availability Statement:** The Fermi-LAT spectrum of the blazar 1ES0229+200 resulting from our analysis will be shared on reasonable request to the corresponding author.

**Acknowledgments:** We acknowledge helpful discussions with our student Y. Verminskaya. All graphs in the present paper were produced with the ROOT software toolkit [68]. E.P. thanks the Foundation for the Advancement of Theoretical Physics and Mathematics "BASIS" (Contract No. 20-2-10-7-1) and the Non-profit Foundation for the Advancement of Science and Education "INTELLECT" for the student scholarships.

**Conflicts of Interest:** Authors declare no conflict of interest. The funders had no role in the design of the study; in the collection, analyses, or interpretation of data; in the writing of the manuscript, or in the decision to publish the results.

## Notes

1    https://github.com/fermi-lat/Fermitools-conda; accessed on 20 September 2020.
2    https://github.com/fermiPy/fermipy; accessed on 20 September 2020.

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
