# Peer review of "Nonthermal Radiation of the Extreme TeV Blazar 1ES 0229+200 from Electromagnetic Cascades on Infrared Photon Field"

_universe, doi:10.3390/universe7120494_

Round 1
Reviewer 1 Report
The paper entitled "Nonthermal Radiation of the Extreme TeV Blazar 1ES 0229+200 from Electromagnetic Cascades on Infrared Photon Field" by Dzhatdoev et al. presents a novel interpretation of the SED of extreme TeV blazars. The novel idea applied to the SED fitting involving the IR emission of the host galaxy is worth exploring. The paper is well-written, clear and to the point and I'd have no difficulty to recommend the paper for acceptance once one source of concern is addressed.
* l. 134: The ultraviolet component is likely due to starlight from the bulge of the host galaxy.
I am not convinced by this statement. The elliptical host template already includes an UV excess. In addition, the SED shape beyond 10 eV is understandably poorly constrained. The UV excess fitted with a black body is a rather ad hoc solution. The authors might consider: (1) actual fits of the UV excess in elliptical galaxies to check about the plausibility of their fit. (2) At least discuss the possibility of thermal emission from an accretion disk.
* Repeat in the caption of Fig. 3 that the curve corrected for redshift effect is actually due to the optical depth of the PP process.
Author Response
We are grateful to the anonymous reviewer for comments and proposed corrections. We address all concerns raised by the reviewer below. In the following text the comments of the reviewer are marked as R, in quotes, and our replies are marked as A.
1. R: “The elliptical host template already includes an UV excess. In addition, the SED shape beyond 10 eV is understandably poorly constrained. The UV excess fitted with a black body is a rather ad hoc solution. The authors might consider: (1) actual fits of the UV excess in elliptical galaxies to check about the plausibility of their fit. (2) At least discuss the possibility of thermal emission from an accretion disk.”
The classical model of the “UV excess” suggests that it “is a stellar phenomenon (as opposed to nuclear activity, for example)”, please see [O’Connell, R.W., Annual Review of Astronomy and Astrophysics 1999, 37, 603–648] (this reference was added into the new version of our paper). A contribution of the accretion flow is also sometimes possible [Cappellari M. et al., The Astrophysical Journal 1999, 519, 117–133] (this reference was also added by us).
Given that “the most striking feature of the UVX [UV excess] phenomenon is its large variation from object to object”; “the amplitude of the UVX in the centers of bright E-Sb galaxies varies from ∼ 4.5 to ∼ 2.0, which is a factor of 10 in the ratio of far-UV to visible flux” [O’Connell, 1999], it is hard to predict the amplitude of the UV excess in a particular source. Therefore, we believe that a more detailed discussion of the nature of the UV excess in the spectra of extreme TeV blazars belongs to a special paper.
We corrected the text of our paper discussing the “UV excess” as follows:
“The ultraviolet component may represent a stellar phenomenon [OConnell1999]. In principle, some contribution to the UV component from an accretion flow is also possible.\footnote{this was the case of the elliptical galaxy NGC 4552 [Cappellari1999]”.
2. R: “Repeat in the caption of Fig. 3 that the curve corrected for redshift effect is actually due to the optical depth of the PP process.”
We added to the caption that the observable gamma-ray component includes the redshift effect and the EBL absorption.
As well, we tried to incorporate into the text of our paper other changes requested by other reviewers.
Reviewer 2 Report
The authors address the issue of a new subclass of "extreme TeV blazars" or ETBs. These sources are challenging the usual synchrotron-self-Compton scenarios and are an excellent opportunity to develop innovative models such as the one presented by the authors.
The authors apply a model of non-thermal emission from leptonic cascades over an external infrared thermal field for the blazar 1ES 0229+200.
Their results show that the gamma-ray spectrum of 1ES 0229+200 is in excellent agreement with the universal spectral shape expected from cascades-originating gamma rays.
The study addresses an important topic and their result regarding the gamma-ray spectrum of 1ES 0229+200 is outstanding. This should be published after, however, properly addressing the major concerns and minor comments expressed below.
Major comment:
- The data analysis part lack details about observing periods, software used, and correctly including upper limits (more details below)
- The VERITAS dataset should not be normalized to HESS, but used as it is.
- The model presented looks quite simplistic with only a few free parameters. Simplicity is often a strength to test models.
However there are numerous underlying assumptions that need to be discussed by the authors, and also general checks on the validity of their model should be performed. Such as:
- SSC emission should be naturally produced as soon as we have ultra-relativistic electrons and synchrotron emission, how do the authors consider it can be neglected?
- What is the geometry of the model? Where is situated the non-thermal emission zone within the IR and UV thermal fields?
- Are these fields isotropic?
- What is the associated temperature and luminosity of these fields?
- The authors should explain why they neglect the UV Blackbody in both the pair absorption and cascade creation.
- What is the energetics of the model compared to the Eddington luminosity?
- What is the equipartition Ub/Ue of their model? Is it better than standard SSC models?
- What is the particle density considered?
- It is a bold assumption of not considering any jet bulk Lorentz factor since it is a fundamental property of blazars. The implication of a non-relativistic blazar jet should be discussed.
Below are more focused comments:
Abstract:
l2: "their intrinsic spectral energy distributions (SEDs) peaked at an energy in excess of 1 TeV"
"Intrinsic" could be understood as "in the jet frame", instead of "non EBL absorbed". I suggest explicitly mentioning what you mean, such as "their spectral energy distributions (SEDs) peaked at an energy in excess of 1 TeV without considering the EBL absorption"
The term intrinsic can be kept for the rest of the text since it is properly described in line 33.
L4: "extremely high" -> higher
l6: "reasonable" is open to interpretation, I suggest "standard"
Main body:
l18 "jets presumably pointing towards the observer"
After several decades of MWL observations and modeling, this can be considered a fact. However most of the jets are not strictly pointing toward the observer, it will be more accurate saying "jets pointing in the close direction to the observer's line of sight"
l37: "with SEDs" -> "with intrinsic SEDs"
l53 "Compton" -> "inverse-Compton"
l61: "the SED" -> "the intrinsic SED"
l65: remove "As a rule,"
l80: "Ku = Up/UB ≫ 1". If Up >> UB as mentioned in the text, Ku should be very small. Please correct.
l98, full paragraph:
One should note that the high gamma_min is indeed specifically important for extreme blazars, but this is an issue affecting most of the observed blazars from all classes.
In the same way, the majority of blazars models from FSRQs to HBLs are widely particle-dominated, this is not a specificity of ETB.
l114: "To produce hard (γ < 2) observable spectrum". The character γ here is confusing since it was not introduced before as a spectral index. -> "To produce hard (spectral index < 2) observable spectrum"
l115: ", bulk motion of a blob "-> ", the bulk motion of a blob"
l131: Authors should provide the temperature and luminosity associated with these two black bodies.
l134: I believe the starlight from the bulge of the host galaxy (or star formation) is already included in the template as the peak at 10 eV (unless this galaxy is known to have an unusually high star formation rate). The observed UV peak seems to match perfectly well the expected "big-blue-bump" of the accretion disk seen in many blazars. The authors should discuss why they favor the scenario of excessive star formation instead of the more common one of the accretion disk signature.
l137: "while observations reveal a more energetic component peaked at ≈ 6 · 10 − 2 eV."
The lowest energy points seem to actually be an upper limit (from Biteau 2020). Upper limits should be clearly identifiable in your plot. So, considering a UL, the observations do not show any evidence of a component peaked at 6e-2 eV.
This reduces the relevance of having a strong IR blackbody at this frequency since only the second IR data point shows an excess above the host galaxy template.
This excess can naturally be explained by synchrotron emission (compact or extended), this should be discussed. It actually seems that the low-energy extrapolation of the X-ray component could naturally match the observed excess (when added to the host emission) if extrapolated down to IR.
Figure 1: Multiple datasets are presented with different markers on the plot. There should be a legend associated with each marker.
l152: "The VERITAS spectrum was slightly re-normalized to better fit the H.E.S.S. spectrum."
Why do the authors want the VERITAS spectrum to fit the HESS spectrum? These are two individual measurements. Unless strongly justified, it is very unusual to modify a-posteriori an analyzed dataset.
I strongly recommend using the original VERITAS spectrum, without modification.
l153: "We have performed a Fermi-LAT [24] data analysis for 1ES 0229+200 over the time period of 2008-08-04 – 2020-09-20 and the energy range of 1 GeV – 500 GeV following the standard recommendations for a point-like source spectral analysis issued by the Fermi-LAT collaboration"
For the sake of data reproducibility, the authors should give more details such as the version of the software fermipy/fermitools, the version of galactic and isotropic background models, and the underlying point source catalog used for the spectral source models in the Region of interest.
l157: "The last and the third from the last energy bins yielded only upper limits which are not shown in the figure"
Upper limits still contain relevant statistical information and should be shown, unless outside the boundaries of the plot. If the authors want to avoid having upper limits, I suggest they use larger energy binning in their Fermi analysis.
Only the time period of the Fermi dataset is mentioned. The observing periods of other instruments should be presented as well. It is mentioned that 1ES 0229+200 is variable on a yearly timescale. How does this variation affects the SED shape given that the datasets are not simultaneous?
Figure3 and 4: It is difficult to recognize VERITAS from HESS data, please use two different marker colors.
Figure 4: The black dashed segment should be mentioned in the caption
l187: B = 0.9 mG.
This looks quite low compared to usual blazar models, so I guess the emission zone is extremely far from equipartition (particle dominated) as the other models. This should be discussed.
It is not clear if the SED is fitted "by eye" or using a fitting algorithm, it should be mentioned.
Author Response
We are grateful to the anonymous reviewer for comments and proposed corrections. We address all concerns raised by the reviewer below. In the following text the comments of the reviewer are marked as R, in quotes, and our replies are marked as A.
1. R: “The data analysis part lack details about observing periods, software used, and correctly including upper limits (more details below)”
A: Our own analysis concerns only Fermi-LAT data. The Fermi-LAT observation period of 2008-08-04 – 2020-09-20 was indicated in the text of our paper in the submitted version (please see Subsection 5.1).
Concerning the Fermi-LAT software, we include the following sentence: “The Fermitools software (version 1.2.23) as well as the fermiPy package [27] (version 0.19.0) 8 were utilized for this analysis” with references to Fermitools and fermiPy.
Concerning other datasets, we refer the reader to papers of H.E.S.S., VERITAS, and the paper [Biteau, J. et al.. Nature Astronomy 2020, 4, 124–131]:
“For details concerning the H.E.S.S. and VERITAS data analysis the reader is referred to [25] and [11], respectively; for details concerning all datasets except the H.E.S.S., VERITAS, and Fermi-LAT ones — to [10].”
All measurements of the SED of 1ES 0229+200 except the Fermi-LAT, H.E.S.S., and VERITAS ones, were presented in [Biteau, J. et al.. Nature Astronomy 2020, 4, 124–131] and made publicly available.
The H.E.S.S. paper on 1ES 0229+200 spectral measurements cited by us [Aharonian, F. et al., Astronomy & Astrophysics 2007, 475, L9–L13] does not include any upper limits. The VERITAS and Fermi-LAT upper limits are now included to Fig. 3.
2. R: “The VERITAS dataset should not be normalized to HESS, but used as it is.”
A: This is corrected.
3. R: “The model presented looks quite simplistic with only a few free parameters. Simplicity is often a strength to test models. However there are numerous underlying assumptions that need to be discussed by the authors, and also general checks on the validity of their model should be performed.”
A: The purpose of this paper is mainly to present the basic idea: “This paper is meant to be the first in series of works aimed at the understanding the nature of extreme TeV blazars. Therefore, we leave some subjects for future study”. It is hardly possible to discuss all caveats/validity checks in a single paper, much less so in ten days given by the Editors to the authors of this paper to prepare the revised version. However, some of these are dealt with in Subsection 6.4 (this Subsect. is newly introduced to the text).
4. R: “SSC emission should be naturally produced as soon as we have ultra-relativistic electrons and synchrotron emission, how do the authors consider it can be neglected?”
A: We estimate the typical optical depth for the inverse Compton process on synchrotron photons. Compared to the IR photons, the synchrotron photons have relatively high energy, but they may have much lower density. An order-of-unity estimate results in the optical depth for the IC scattering on the synchrotron photons ~ 0.01. Of course, we do not claim that the SSC component could be neglected in all cases. We merely show that this component could be subdominant.
5. R: “What is the geometry of the model? Where is situated the non-thermal emission zone within the IR and UV thermal fields?”
A: Most probably, the spatial range in question is between 0.01 pc and 100 pc (please see the last paragraph of Sect. 4). So far, we are not able to obtain any strong constraints on this spatial scale. This is natural since the model is “linear” (the photon field is not created by primary high energy particles, as opposed to the case of the SSC model). To provide such constraints, some important new pieces of information are required: a) measurements or at least strong constraints below 0.05 eV, b) reliable information about the angular extent (angular radius) of the jet, c) detailed study of the parameter space with the aim to exclude the values of E_b below 0.05 eV. We have a strong opinion that this study belongs to a separate work and separate publication.
6. R: “Are these fields isotropic?”
A: “For simplicity we assume that the IR photon field is isotropic.” (the last sentence inserted to Sect. 4). However, even anisotropic photon fields would not ruin the proposed concept. However, the technical difficulties of accounting for such anisotropic photon fields could be considerable.
7. R: “What is the associated temperature and luminosity of these fields?”
A: In principle, these IR photon fields could even have a non-thermal nature from “relic” electrons that escaped the jet. In this case the notion of temperature is somewhat misleading. We assumed the typical energy of IR photons E_b= 0.06 eV (please see Subsect. 5.4, Table 1). The values between 0.01 eV and 0.1 eV are reasonable. To exclude some values of E_b, a detailed study of the parameter space is necessary.
8. R: “The authors should explain why they neglect the UV Blackbody in both the pair absorption and cascade creation.”
A: The classical model of the UV component (“UV excess”) in elliptical galaxies suggests that it “is a stellar phenomenon (as opposed to nuclear activity, for example)”, please see [O’Connell, R.W., Annual Review of Astronomy and Astrophysics 1999, 37, 603–648] (this reference was added into the new version of our paper, please see Sect. 4). “Assuming that the UV component of photon field represents a stellar phenomenon on ∼ kpc scale (see Section 4), the pair production depth on this UV excess is small ( ≪ 1).” (added to Subsection 6.4).
9. R: “What is the energetics of the model compared to the Eddington luminosity?; What is the particle density considered?”
A: These quantities could not be calculated without reliable information about the angular extent (angular radius) of the jet.
10. “What is the equipartition Ub/Ue of their model? Is it better than standard SSC models?”
This parameter is not relevant in the framework of the present model, because there is no requirement that the particles are confined in blobs (as is usually required in more conventional models).
11. R: “It is a bold assumption of not considering any jet bulk Lorentz factor since it is a fundamental property of blazars. The implication of a non-relativistic blazar jet should be discussed.”
A: “In the present paper we do not invoke bulk motion of matter in blobs/jet to explain the non-thermal radiation spectrum of 1ES 0229+200. In this respect, our model is similar to the one presented in [28] for the case of the radiogalaxy M87.” (added to Subsect. 6.4)
Our model is not the first time when the bulk motion of matter in blobs/jet was not invoked. The reader is referred to [Neronov, A.; Aharonian, F.A. The Astrophysical Journal 2007, 671, 85–96] for more details.
12. R: “l2: "their intrinsic spectral energy distributions (SEDs) peaked at an energy in excess of 1 TeV"
"Intrinsic" could be understood as "in the jet frame", instead of "non EBL absorbed". I suggest explicitly mentioning what you mean, such as "their spectral energy distributions (SEDs) peaked at an energy in excess of 1 TeV without considering the EBL absorption"
The term intrinsic can be kept for the rest of the text since it is properly described in line 33.”
A: We prefer the form “compensated for the effect of γ-ray absorption on extragalactic background light photons”.
13. R: “l6: "reasonable" is open to interpretation, I suggest "standard"”
A: We prefer the form: “conventional γ-ray emission models”.
14. R: “l80: "Ku = Up/UB ≫ 1". If Up >> UB as mentioned in the text, Ku should be very small. Please correct.”
A: If Up >> UB, then Up/UB ≫ 1.
15. R: “l98, full paragraph:
One should note that the high gamma_min is indeed specifically important for extreme blazars, but this is an issue affecting most of the observed blazars from all classes.
In the same way, the majority of blazars models from FSRQs to HBLs are widely particle-dominated, this is not a specificity of ETB.”
A: The notion of “particle dominance” is an interpretation. For other classes of blazars, there are other remedies. For instance, for the neutrino-emitting blazar TXS 0506+056 (one of the severest cases) there is no “particle dominance” if one assumes a model where high energy gamma-rays were produced as a result of synchrotron radiation of “hadronic electrons” [as was shown by us at the TeVPA-2019 conference]. Please let us consider these problems in separate paper(s).
16. R: “l134: I believe the starlight from the bulge of the host galaxy (or star formation) is already included in the template as the peak at 10 eV (unless this galaxy is known to have an unusually high star formation rate). The observed UV peak seems to match perfectly well the expected "big-blue-bump" of the accretion disk seen in many blazars. The authors should discuss why they favor the scenario of excessive star formation instead of the more common one of the accretion disk signature.”
A: “It is believed that the accretion flow in Bl Lacs is radiatively inefficient, with high tempera-
ture beyond the UV energy range. Therefore, it is not likely that the UV component is dominated by the accretion flow.” (added to Sect. 4).
17. R: “This [IR] excess can naturally be explained by synchrotron emission (compact or extended), this should be discussed. It actually seems that the low-energy extrapolation of the X-ray component could naturally match the observed excess (when added to the host emission) if extrapolated down to IR.”
A: This is indeed possible, especially if one allows for an additional population of (relatively low-energy) electrons that have escaped the emission zone, but are still situated not far from it. “The blackbody fits of the UV and IR presented in Figure 1 are formal ones and do not have direct physical meaning. In particular, the IR component could be in part produced as the result of synchrotron radiation of electrons.” (added to Subsect 6.4).
18. R: “Figure 1: Multiple datasets are presented with different markers on the plot. There should be a legend associated with each marker.”
A: We appreciate this comment very much, but isn’t it possible to deter this to the production stage given that this issue is a purely typographic one?
19. R: “It is mentioned that 1ES 0229+200 is variable on a yearly timescale. How does this variation affects the SED shape given that the datasets are not simultaneous?”
A: This is the subject of a future work. Besides, to the best of our knowledge, the fact that the observations are not strictly simultaneous never precluded the publication of results/models in blazar physics, even for the case of highly variable "classical" BL Lacs or FSRQs, much less in the case of extreme blazars.
20. R: ““l187: B = 0.9 mG.
This looks quite low compared to usual blazar models, so I guess the emission zone is extremely far from equipartition (particle dominated) as the other models. This should be discussed.””
A: [Böttcher, M.; Dermer, C.D.; Finke, J.D. The Astrophysical Journal 2008, 679, L9–L12] cited by us have B ~ 10 \mu G. Besides, “the physical meaning of B is the strength of the turbulent component of magnetic field. The regular magnetic field along which the particles move could have a significantly higher strength than B” (please see Subsect 5.4). The notion of “particle dominance” does not have any immediate physical meaning if confinement of particles in blobs/jet is not required. This was discussed above.
21. R: “It is not clear if the SED is fitted "by eye" or using a fitting algorithm, it should be mentioned.”
A: “This fit was obtained without any formal optimization with a dedicated algorithm, but just by choosing reasonable values of input parameters and performing several tries.” (added to Subsect 5.3).
Other comments of the reviewer were taken into account. As well, we tried to incorporate into the text of our paper other changes requested by other reviewers.
Reviewer 3 Report
Please check the attached file.

Author Response
We are grateful to the anonymous reviewer for comments and proposed corrections. We address all concerns raised by the reviewer below. In the following text the comments of the reviewer are marked as R, in quotes, and our replies are marked as A. In total, we have received more than 50 comments/suggestions from three reviewers. It is not possible to implement them all considering the very short response time (ten days) required from the authors by the Editors.
1. R: “Line 1: The names and acronyms usually seen in the literature are either full name “extreme blazars” (as in, e.g., Biteau et al. 2020, which is cited several times in this work), or acronym “EHBL” (e.g. TeVCat). Probably EHPS could also be applicable, but the authors accepted the classification according to the peak position in the gamma-ray band, so EHPS is less favourable. I honestly do not see the need for another name or acronym, such as “ETB” introduced by the authors. Indeed, introducing alternative names and acronyms for the same phenomenon contributes only to confusion. Moreover, “extreme TeV blazars” is somewhat misleading because the characteristic of this type of sources is the peak position at energies above 1 TeV, while the name could imply extreme flux or extremely variable flux (or some other characteristic for that matter) in TeV energy range, none of which is necessarily true.”
A: EHBL and ETB is not the same thing. EHBL is usually defined by the frequency of the lower-energy peak. Besides, the notion of “extreme TeV blazar” is not that rare in the literature:
https://ui.adsabs.harvard.edu/abs/2021MNRAS.505.1940K/abstract
https://ui.adsabs.harvard.edu/abs/2017A%26A...603A..59D/abstract
https://ui.adsabs.harvard.edu/abs/2008AIPC.1085..447S/abstract
https://ui.adsabs.harvard.edu/abs/2020MNRAS.496.1430P/abstract
https://ui.adsabs.harvard.edu/abs/2021A%26A...654A..96Z/abstract
https://ui.adsabs.harvard.edu/abs/2014A%26A...568A.110O/abstract
https://ui.adsabs.harvard.edu/abs/2012ApJ...749...63M/abstract
Given that the field is extremely young, it is an impressive record. To the best of our knowledge, this have never caused any problems given that the authors explain the notions. We beg the permission to use the term “extreme TeV blazar” and the acronym ETB for brevity.
2. R: “Line 24: In the sense of short variability timescales, the paper by Aleksić et al. (2014, https://ui.adsabs.harvard.edu/abs/2014Sci...346.1080A/abstract) can be considered. Granted the source in case (IC 310) is a radio galaxy, but still with a minute timescale variability. More importantly, one of the explanations for the observed emission considered by Aleksić et al. includes acceleration of particles within the SMBH magnetosphere, not unlike what authors presented here.”
A: We fully agree that IC 310 could be relevant for the proposed scenario. The reference to this paper is added to the end of Subsect. 6.2.
3. Line 39: In “By the end of 2019 about ten ETBs were discovered.”, a reference to TeVCat (http://tevcat2.uchicago.edu/) or a similar source would be useful.
A: In fact, the table of EHBLs/ETBs is available in Biteau et al. (2020). This is corrected (another reference to this paper is included).
4. "Lines 65-66: I would not be so confident and claim this to be “a rule”. With only ~10 sources detected so far, and given the flux and instrument sensitivity in energy ranges above 1 TeV, I think the most that we can say is that so far (with the exception of a day-scale flare of 1ES 1218+304) strong or fast flux variability has not been observed so far."
A: A similar change was proposed by another referee (however, we are not sure whether he/she believes that this is more than a “rule” rather than less than a “rule”). This is corrected; we simply state what was observed without using the word “rule”: “ETBs do not reveal fast (day-scale or even week-scale) or strong (flux change by an order of magnitude or more) γ-ray variability”.
5. R: “Line 134: Is there a reference to support this claim?”
A: The classical model of the “UV excess” suggests that it “is a stellar phenomenon (as opposed to nuclear activity, for example)”, please see [O’Connell, R.W., Annual Review of Astronomy and Astrophysics 1999, 37, 603–648] (this reference was added into the new version of our paper). A contribution of the accretion flow is also sometimes possible [Cappellari M. et al., The Astrophysical Journal 1999, 519, 117–133] (this reference was also added by us).
6. R: “Line 152: The statement that “The VERITAS spectrum was slightly re-normalized...”
demands a more detailed explanation and description. How much is “slightly”? How exactly was the renormalisation done, on what grounds, and what did it include? Theway this is written now, it can include all sorts of sins, and it should be described in more detail to avoid any suspicion. If the procedure is somewhat lengthy, it can be placed in an appendix.”
A: In the new version of the paper, we use the VERITAS dataset without any re-normalization.
7. R: “Line 158: Why were the upper limits from LAT not shown in the SED? Upper limits can have a discriminating role on model applicability.”
A: This is now corrected, the Fermi-LAT and VERITAS upper limits are included. The H.E.S.S. paper used by us does not include any upper limits.
8. R: “Lines 161-163: It is not clear here which are the target photons for the inverse Compton process. Probably IR photons, but it should be specified.”
A: This is already explained in the last paragraph of Sect. 4: “In this case these IR photons could serve as a target for high-energy electrons that could produce VHE γ rays via the IC process. The same electrons could produce the X-ray component of the spectrum via the synchrotron process. Let us see what shape of the observable spectrum we could expect in the framework of this scenario.”
9. R: “In relation to the previous point: Are synchrotron photons considered as possible targets for inverse Compton? If not why not, and if yes, what is their contribution to the total SED?”
A: This is now explained in the first paragraph of (new) Subsect 6.4. We show that in some cases the contribution of the SSC component is subdominant. We do not claim that it is always subdominant.
10. R: “Figure 3 caption: Is the blue dashed curve obtained with 100 TeV, or with 300 TeV primary electrons?”
A: This is for the case of E_{me} = 100 TeV. The text was corrected accordingly.
11. R: “Figure 3 caption: What does “including the redshift effect” mean?”
A: This is changed to “including redshift”, i.e. changing the gamma-ray energy E_gamma→ E_gamma(1+z). We are pretty sure that most readers will understand the meaning.
12. R: “There is something not clear to me about the intrinsic gamma-ray horizon and the intrinsic optical depth for gamma rays. How do you define each of these quantities, and which one do you consider to be a more fundamental one? Furthermore, τ int is parameterized as E/E a , and E a is defined through Eq. (4), but there is no explanation what E t is. You only state that it was set to 1TeV, without explaining what it represents, and why this value was chosen. Later, in line 223, you state that the value of E t is defined by the physics of the PP process, but you still do not say what it is and how it’s value was determined (BTW “determined” in the sense of line 223 is probably more suited than “defined”). Similarly for ε. You also state in lines 224-225 that “these parameters are in fact defined by the values of Eb, B, and Kcs.”, but you don’t say how they are defined. I suggest adding an appendix with these “less important” definitions, and in the main text explaining what you consider (or how you
define) (i) “intrinsic gamma-ray horizon”, (ii) “intrinsic optical depth for gamma rays”, (iii) what E t represents, (iv) on what grounds you chose E t = 1 TeV. I find this the most important major remark to the manuscript.”
A: “The intrinsic optical depth for γ rays (more precisely, its dependence on the energy τ int ( E ) ) is usually calculated by integrating the interaction rate over the distance. The definition of the interaction rate could be found e.g. in [33]. The energy of the γ-ray horizon is the energy corresponding to τ int ( E ) = 1.” (added to the text). Thus, the dependence of the optical depth on the energy is the more “fundamental” quantity; the energy of the horizon could be calculated from it.
“ E_t is the effective threshold energy of background photon interacting with gamma-ray via the PP process. For instance, the threshold for a 1 TeV gamma-ray is ~1 eV; E_t is inversely proportional to the gamma-ray energy” (added to the text).
Concerning epsilon, for the sake of simplicity it could be regarded as another fitting parameter. How, in principle, one could calculate the value of epsilon? The answer is very simple: one could run a full Monte-Carlo simulation, then fit the resulting spectrum with our eq. (1), and obtain the value of epsilon.
13. R: “Line 209: What exactly does this statement mean? Does it mean that Ea/Exc < 10 is independent of the electron spectrum? If so, I suggest that you state that explicitly.”
A: We just described what we assumed and what we get: namely, the result for this particular set of assumptions/parameters yields Ea/Exc < 10 --- the value that is problematic for other models.
14. R: “Lines 216-218: You admit that there is some degeneracy on the combinations of values for the model parameters. On what grounds did you choose the values cited in Table 1, and why not some other combination of values?”
A: “This fit was obtained without any formal optimization with a dedicated algorithm, but just by choosing reasonable values of input parameters and performing several tries.” (added to the description of Fig. 3 in the main text). Of course, other values are possible. However, we believe that the priority should be given to the improvement of the model rather that formal matters such as making a huge numbers of tries. In the present paper our main aim is to discuss ideas. We are preparing a more detailed paper with more formal results that will be submitted to other journal that allows the authors to take more time to correct their papers.
15. R: “Lines 238-241: What are the expected (shortest) timescales for flux variability induced by “a variation in the accretion rate” and the consequential processes?”
A: The month timescale variability is in principle possible --- this is clear from tidal disruption events. However, the exact value depends on the type of the accretion flow. A detailed investigation is well beyond the scope of the present paper.
16. R: “Lines 251-252: You mention that primary electrons can be produced as a byproduct of inelastic proton scattering on IR photons. However, this would also generate a high flux of neutrinos. Can you estimate the neutrino flux in this case? Is it something potentially detectable with the IceCube neutrino observatory? The neutrino flux is a potential discriminatory criterion for this scenario.”
A: We appreciate this remark very much. However, in this paper we mainly deal with gamma-rays. The calculation of the neutrino flux is well beyond the scope of the present paper.
17. R: “Line 32: “Neglecting” is probably not the best choice here. “Correcting for”, “Accounting for”, or “Subtracting” would probably fit the meaning better.”
A: We respectfully disagree. Most of the studies neglect the contribution of intergalactic cascades outright. This is discussed in great details in our paper [Dzhatdoev, T.A.; Khalikov, E.V.; Kircheva, A.P.; Lyukshin, A.A. Astronomy & Astrophysics 2017, 603, A59] which we cite. We beg the reviewer to have a look.
Other comments of the reviewer were taken into account. As well, we tried to incorporate into the text of our paper other changes requested by other reviewers.
Round 2
Reviewer 1 Report
The authors have addressed my concerns and attended to the suggestions of the second reviewer.
Author Response
We are grateful to the anonymous reviewer for comments and proposed corrections. We address all concerns raised by the reviewer below. In the following text the comments of the reviewer are marked as R, in quotes, and our replies are marked as A.
1. R: “It appears that the HESS spectrum has changed but not the VERITAS one. I assume the authors previously normalized HESS instead of VERITAS as specified in the first paper version. This should be fine without any requested change on the paper, I just want confirmation or clarification from the authors.”
A: Actually, it were the VERITAS points (red circles) that changed. They fell somewhat below their “old” values. For instance, in the “old” version the first (lowest-energy) red circle had the value that is somewhat larger than 10^{-12} [TeV/(cm^{2}*s)], while in the new version it is almost equal to 10^{-12} [TeV/(cm^{2}*s)].
2. R: “l306: The authors should describe how they obtain the value of 400 between synchrotron and IR photon density. The Synchrotron field span over a much larger energy range than the IR field, it is not clear how the SSC emission can be deduced from the IR photon-photon opacity.”
A: We calculated the ratio of the number density for the IR photons to the X-ray photons K_{IR − X}. If the X-ray photons were isotropic, than this ratio would be around 10^{3}. We obtained this by integrating over the spectra (accounting for the fact that the X-ray photon field could span a larger energy range than the IR photon field). However, the IR photon field could be isotropic, while the X-ray photon field is anisotropic. We missed this factor at the previous round of the review, now this oversight is corrected. Assuming the jet with the angular radius theta_{jet} of 0.1 rad, we get K_{IR − X}~ 10^{5}, i.e. the SSC component is subdominant. We note that this is the case of "purely geometrical beaming" ~theta_{jet}^{2} because we did not introduce blobs / bulk motion, only particles that propagate along the jet. The text is corrected accordingly. Considering the large factor theta_{jet}^{2}, the "finesses" with the ratio of tau_{IC}/tau_{/gamma/gamma} are now unnecessary; these were removed.
3. R: “It is generally true for HBLs but thermal accretion disk signatures are present in most of the other blazar classes, including IBLs and LBLs. Saying this it is usually assumed that only the central part of the disk is inefficient for the weaker sources, an outer thermal disk emission is still expected (such as presented in Narayan 1996: https://ui.adsabs.harvard.edu/abs/1996ApJ...457..821N/abstract). Without observational proof of excess in star formation, the authors should mention that both options are theoretically possible and should be kept in mind.”
A: This is corrected: “The ultraviolet component may represent a stellar phenomenon [28]. In principle, some contribution to the UV component from an accretion flow is also possible. It is believed that the inner part of the accretion flow in Bl Lacs is radiatively inefficient, with high temperature beyond the UV energy range. However, as discussed in [30], the outer part of the accretion flow could represent a standard Shakura-Sunyaev accretion disk [31].”
Thus, we acknowledge that some contribution to the UV component from the accretion flow is possible.
4. R: (About the legend to Fig. 1) “This comment is a minor one that should not impact the general review recommendation. However, it should be addressed anyway before publication.”
A: This is corrected, the necessary references added.
5. R: “The dataset does not need to be simultaneous if the authors show that the source can be considered as steady (or that the variability amplitude is weak enough to not impact their modeling results) between the multiple observing periods considered. This should be mentioned in the text. This issue can actually be quite critical for sources with high variability. Mixing various activity states can lead to wrong and misleading SED features that do not exist and make modeling results obsolete. This is an issue one should always keep in mind when working on multi-instrument datasets from variable sources.”
A: This is corrected: “We note that the observations with different instruments were not strictly simultaneous. However, as the source under study does not display a very strong or fast variability, in what follows we use the datasets resulting from these observations.”
6. We added the redshift of the studied source (z= 0.14) with the necessary reference.
7. In the “old” version there was an error concerning the IR and UV luminosities (incorrect z= 0.186 instead of 0.14 was assumed). Now this is corrected: 8·10^{43} [erg/s] and 2·10^{44} [erg/s] → 4·10^{43} [erg/s] and 1.2·10^{44} [erg/s], respectively.
Reviewer 2 Report
Please find the review in the attached pdf. The new comments are highlighted in green.

Author Response

(The authors gave the same response as above.)
